# Application of Palliative Hemostatic Radiotherapy in Canine Unresectable Oral Melanoma: A Case Report

**DOI:** 10.3390/ani14121746

**Published:** 2024-06-09

**Authors:** Myounghun Kim, Inseong Jeong, Gijong Lee, Cheol Park, Youngwhan Kim, Kidong Eom, Jaehwan Kim

**Affiliations:** 1Department of Veterinary Medical Imaging, College of Veterinary Medicine, Konkuk University, Seoul 05029, Republic of Korea; atles5417@konkuk.ac.kr (M.K.);; 2Royal Animal Medical Center, Seoul 02140, Republic of Korea

**Keywords:** hemostatic radiotherapy, hemorrhage, palliative radiotherapy, oral melanoma, dog

## Abstract

**Simple Summary:**

Hemorrhage can pose a life-threatening risk in patients with advanced tumors, prompting the use of palliative radiotherapy for hemostasis in humans. Oral melanoma, which is commonly observed in dogs, frequently presents with a hemorrhage. This case report describes the use of palliative radiotherapy to treat a chronic hemorrhage resulting in severe anemia in a dog with oral melanoma. Following the radiotherapy, hemorrhage cessation and recovery from the anemia were noted. Moreover, a reduction in tumor size, pain, and inflammation, along with increased food intake, have been observed to contribute to enhanced quality of life.

**Abstract:**

A 9-year-old castrated male Schnauzer dog, weighing 11.6 kg, presented with a persistent hemorrhagic oral mass. An oral examination revealed a right maxillary oral mass characterized by continuous bleeding, halitosis, and severe pain. A cytological examination led to a provisional diagnosis of malignant melanoma, and, despite the option of aggressive surgery, the owner declined. The blood analysis indicated severe hemorrhagic anemia (hematocrit, 18.2%) requiring a blood transfusion. The patient underwent volumetric modulated arc therapy (VMAT) as part of a palliative radiation protocol, receiving six fractions of 6 Gy weekly for hemostasis and clinical improvement. The hemorrhaging ceased after the second fraction, with a subsequent rise in the hematocrit levels and the resolution of the anemia. Additionally, the intake increased following the second fraction, and effective pain management was achieved in the fourth fraction. Following the last fraction, computed tomography revealed a 20% reduction in the tumor size. This case highlights the potential use of radiotherapy for hemostasis in cases of inoperable hemorrhagic oral melanoma and represents the first report on the application of hemostatic radiotherapy in dogs.

## 1. Introduction

Hemorrhage is a frequent complication in patients with advanced cancer, often leading to oncological emergencies and posing life-threatening risks [1]. The management options for hemorrhages encompass a spectrum of interventions, including dressings, hemostatic agents, and interventional procedures [2]. Palliative radiotherapy has emerged as a standard modality for achieving hemostasis in humans, demonstrating successful outcomes in addressing bleeding tumors across diverse anatomical sites, such as the respiratory, digestive, and reproductive systems, with the reported hemostatic efficacies ranging from 73 to 100% [1,3,4,5,6].

Malignant melanoma is the predominant oral tumor in canines and frequently manifests with symptoms such as halitosis, excessive drooling, facial swelling, pain, dysphagia, and hemorrhage [7]. Characterized by its aggressive nature and propensity for invasion and metastasis, oral melanoma presents challenges for local management. While surgical resection or radiotherapy may be considered for localized control, the invasive nature of the disease often necessitates a partial mandibulectomy or maxillectomy, which are complex procedures [7,8,9]. Consequently, radiotherapy plays a pivotal role in the management of patients with oral melanoma and often serves as a nonsurgical alternative for both palliative and curative objectives [9,10,11,12].

This case report describes a patient with severe anemia resulting from a chronic hemorrhage due to oral melanoma, wherein hemostatic radiotherapy was utilized for palliative purposes to manage the hemorrhage effectively.

## 2. Case Presentation

A 9-year-old castrated male Schnauzer weighing 11.6 kg presented with a 1-month history of oral hemorrhage, facial swelling, and halitosis. Oral examination revealed a black mass accompanied by a hemorrhage in the right maxilla. Samples were obtained using fine-needle aspiration. On oral palpation, the pain response was pronounced (Colorado State University Veterinary Medical Center Canine Acute Pain Scale 3/4). Despite the administration of various analgesics such as piroxicam (0.3 mg/kg, PO, q24h), gabapentin (10 mg/kg, PO, q24h), and tramadol (3 mg/kg, PO, q24h), effective pain management was not achieved. Additionally, the patient’s food intake was significantly reduced due to the oral tumor. In the cytological examination, immature mesenchymal cells with dark green granules in the cytoplasm were observed. Additionally, prominent multiple nuclei, anisocytosis, anisocytosis, anisokaryosis, and variations in the nucleus-to-cytoplasm ratio were noted. Therefore, the oral mass was provisionally diagnosed as malignant melanoma (Figure 1). A complete blood count (CBC) indicated moderate anemia (hematocrit, 28.6%; reference range: 37–55%) and reticulocytosis (4.26%; reference range: 0.1~2.0%), suggesting a regenerative response in hemorrhagic anemia. Additionally, an increase in mean corpuscular volume (82 fL; reference range: 60–74 fL) and a decrease in mean corpuscular hemoglobin concentration (30.2 g/dL; reference range: 31–36 g/dL) were observed. Serum biochemistry analysis revealed decreased albumin levels (2.1 g/dL, reference range: 2.3–3.9 g/dL) and elevated alkaline phosphatase (679 U/L, reference range: 20–155 U/L), alanine aminotransferase (94 U/L, reference range: 3–50 U/L), aspartate aminotransferase (41 U/L, reference range: 10–37 U/L), amylase (2790 U/L, reference range: 388–1007 U/L), lipase (198 U/L, reference range: 5–90 U/L), and increased C-reactive protein (171 mg/L, reference range: 0–10.0 mg/L).

To evaluate the tumor size, the extent of invasion, and metastasis, 64-multi-slice helical CT scans were conducted using a pre- and post-contrast whole-body computed tomography (CT) scanner (Optima CT660, GE Healthcare, Tokyo, Japan). The patient was positioned in ventral recumbency and secured with a custom-made vacuum cushion (Vac-cushion; Chunsung, Seoul, Republic of Korea). Imaging was conducted under specific parameters: 120 kVp, 160 mA, and 1.25 mm slice thickness. A power injector (MEDRAD^®^ Stellant, Bayer, Indianola, PA, USA) was used to administer 600 mg iodine/kg iohexol (Omnipaque™ 300, GE Healthcare, Shanghai, China). Post-contrast computed tomography (CT) revealed a mass with heterogeneous contrast enhancement, measuring up to 41 mm in maximum diameter within the right maxilla. Additionally, evidence of bone erosion was observed in the maxillary bone, turbinate bone, and hard palate adjacent to the oral mass (Figure 2). Furthermore, the mass had invaded the interior of the right nasal cavity. Although metastasis to the regional lymph nodes and distant organs was not definitively identified, the patient was classified as WHO stage 3/4 through CT.

After computed tomography (CT), the patient experienced persistent bleeding. By the fourth day following the CT scan, the condition had progressed to severe anemia (hematocrit, 18.2%; reference range: 37–55%), necessitating immediate blood transfusion. Although there was a temporary increase in hematocrit to 23.3% by 2 days post-transfusion, it dropped to 19.9% the following day. Ongoing hemorrhage presents challenges in effectively managing the patient’s anemic condition.

Despite the recommendation for surgical excision, the owner declined due to concerns regarding its invasiveness. Instead of surgical removal, palliative radiotherapy was chosen to control hemorrhage and improve quality of life. Volumetric modulated arc therapy (VMAT) was administered using a 6 MV linear accelerator (clinic iX, Varian Medical Systems, Inc., Palo Alto, CA, USA) equipped with a 5 mm leaf width multileaf collimator (MLC). Inverse treatment planning was conducted using the Varian Eclipse treatment planning system (Eclipse™ version 13.7.33, Varian Medical Systems, Inc., Palo Alto, CA, USA) utilizing two 360° volumetric arcs (Figure 3). The prescribed dose was 36 gray (Gy), delivered in six fractions of 6 Gy each weekly. Given the patient’s severe hemorrhagic anemia before radiotherapy, two treatments were administered in the first week to achieve rapid hemostasis, resulting in a total treatment duration of 5 weeks. The gross tumor volume (GTV) was delineated as the contrast-enhanced lesion observed in CT images. To deliver a dose of 120% at the center of the GTV, a 3 mm isotropic volume within the GTV was designated as the simultaneous integrated boost (SIB). The planning target volume (PTV) included a 3 mm isotropic expansion from the GTV. The dosimetric objectives were aimed at 100% prescription dose coverage of 99% of the GTV and 95% of the PTV (Figure 3). The organs at risk (OAR) included the eyes, lenses, optic nerves, optic chiasm, brain, and palatal mucosa. Quality assurance was performed using gamma analysis with the Varian portal dosimetry system on individual fields, with a minimum of 95% gamma for a 3 mm distance to agreement and a 3% absolute dose difference defined as passing the quality assurance criteria. The volumes of target structures and calculated radiation doses/volumes to those structures are summarized in Table 1.

After the first fraction, the hemorrhage was significantly reduced, and, 2 days later, following the second fraction, the hemorrhage stopped. No instances of additional hemorrhage occurred after hemostasis (Figure 4). The hematocrit, initially 19.9% on the day of the first session, gradually increased. After the second session, it reached 25.1% and increased to 39.3% after the third session. Anemia effectively resolved from the third session onward (Figure 5). After the fourth session, both the mean corpuscular volume (69.7 fL; reference range: 60–74 fL) and the mean corpuscular hemoglobin concentration (33.3 g/dL; reference range: 31–36 g/dL) returned to within their respective reference ranges. Additionally, the C-reactive protein level (1.2 mg/L; reference range: 0–10 mg/L) decreased to within the reference range, indicating an improvement regarding inflammation. Following radiation therapy, the previously unmanaged pain diminished. From the fourth session onward, despite no use of analgesics, oral palpation revealed minimal pain response (Colorado State University Veterinary Medical Center Canine Acute Pain Scale 1/4). In addition, food intake improved from the second session onward. Following the final treatment, a contrast-enhanced CT scan was performed to assess tumor size and metastasis changes. A 20.6% reduction in tumor size compared to the pretreatment size was observed by comparing the sum of the maximum diameters of the tumor (Figure 2). However, metastatic changes were strongly suspected in the right mandibular lymph node and lung, based on CT findings, leading to a worsening of the WHO stage to 4/4.

No instances of radiation toxicity were observed after the radiotherapy. However, on the 84th day after the first fraction, the patient presented with worsening tumor progression and ipsilateral mandibular lymphadenopathy. The patient died at home within 1 week.

## 3. Discussion

This case report describes the rapid hemostasis and subsequent resolution of anemia in a patient with chronic bleeding associated with oral melanoma following palliative radiotherapy using VMAT. In addition to the hemostatic efficacy, the treatment reduced the tumor size, pain management, and malodor.

Palliative radiotherapy aims not to extend the survival time but to improve the quality of life by alleviating the clinical symptoms, including pain, inflammation, and anticipated hemostatic effects [4]. Palliative radiotherapy has been utilized for hemostatic purposes in various hemorrhagic tumors, such as hemoptysis due to pulmonary tumors, gastrointestinal bleeding due to gastric tumors, hematuria due to bladder and prostatic tumors, hematochezia due to colorectal tumors, and vaginal bleeding due to gynecological tumors [1,5,6,13]. While few reports exist in veterinary medicine regarding the resolution of hemorrhages after radiotherapy in cases of adrenal tumors accompanied by hemoperitoneum and reductions in the hemorrhagic discharge in oral squamous cell carcinoma cases, there have been no reports of using radiotherapy for hemostatic purposes [14,15]. While the number of cases in animals where the bleeding was stopped through radiotherapy is limited, the observed similar hemostatic effects to those in humans suggest that palliative radiotherapy may be beneficial for hemostasis in dogs.

Various protocols for hemostatic radiotherapy have been reported in humans [1]. The treatment fraction and biologically effective dose (BED) were compared to determine their effectiveness. In the past, it was believed that 10 fractions and a BED of >39 Gy were required for hemostasis. However, recent reports suggest that effective hemostasis can be achieved with fewer than five fractions and a BED of less than 39 Gy [1,3,13]. Additionally, short treatment courses with fewer than five fractions are preferred because fewer treatment sessions reduce the risk of treatment interruption. Among the protocols that are both short and effective for hemostasis, the shortest protocol involves a single dose of 8 Gy, corresponding to a BED of 14.4 Gy when assuming an α/β ratio of 10 Gy for tumors [3]. In this case, a single dose of 6 Gy did not meet the effective hemostatic dose of the BED of 14.4 Gy, resulting in only a partial reduction in the bleeding. In the initial week, two doses were administered to achieve rapid hemostatic effects, with an additional dose administered 2 days after the first dose, resulting in a BED of 19.2 Gy, exceeding the required BED of 14.4 Gy. As a result, the hemorrhage stopped after the second fraction, demonstrating results similar to those of previous reports, indicating that a BED of 14.4 Gy is necessary for hemostasis. Furthermore, although administering two doses in the first week may have increased the risk of acute radiation toxicity, no acute radiation toxicity was observed. Moreover, as previous reports have not shown significant differences in the treatment efficacy and toxicity between weekly and biweekly protocols, administering two doses in the first week is considered acceptable.

Although the exact mechanism of hemostasis during radiation therapy is not fully understood, several mechanisms have been hypothesized. First, upon radiation exposure, the von Willebrand factor (VWF) is increased from the endothelial cells lining the blood vessels. The released VWF acts as a glue, facilitating the adherence of platelets to the subendothelium of blood vessels. This leads to platelet aggregation and activation, resulting in blood clot formation and hemorrhage cessation [16,17]. Secondly, the radiation exposure triggers inflammation in the endothelial cells, releasing transforming growth factor-β1 (TGF-β1). The released TGF-β1 increases the thrombin production, promotes blood clotting, and activates myofibroblasts, enhancing collagen secretion and vascular fibrosis [18]. Consequently, the blood vessels are sealed, halting the hemorrhage.

There were several limitations in this study. This study focused on the hemostatic effects of palliative radiotherapy in one animal, restricting the generalization of the radiotherapy effects for hemostasis in animals more broadly. Furthermore, the effectiveness of radiotherapy for hemorrhagic tumors in various locations and types remains unverified, necessitating additional further study.

## 4. Conclusions

This case report represents the first application of palliative radiotherapy for hemostatic purposes in a dog. Palliative radiotherapy is known for its analgesic effects and improvement in terms of quality of life but may be considered an effective treatment modality for its potential hemostatic effects as well, particularly in cases of inoperable hemorrhagic oral melanoma.

## Figures and Tables

**Figure 1 animals-14-01746-f001:**
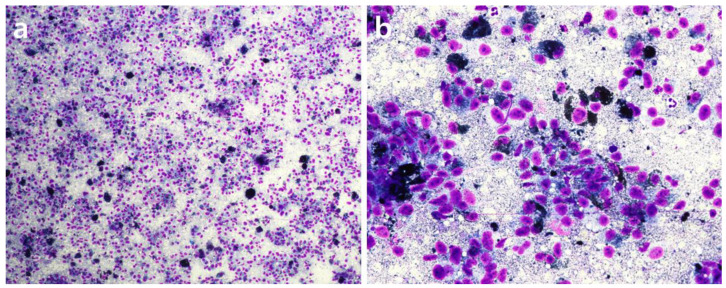
Microscopic appearance of cells obtained through fine needle aspiration of the oral mass ((**a**): 40×; (**b**): 400×). Prominent multiple nuclei, anisocytosis, anisokaryosis, and variations in the nucleus-to-cytoplasm ratio were also noted. These cytological features support a provisional diagnosis of malignant melanoma.

**Figure 2 animals-14-01746-f002:**
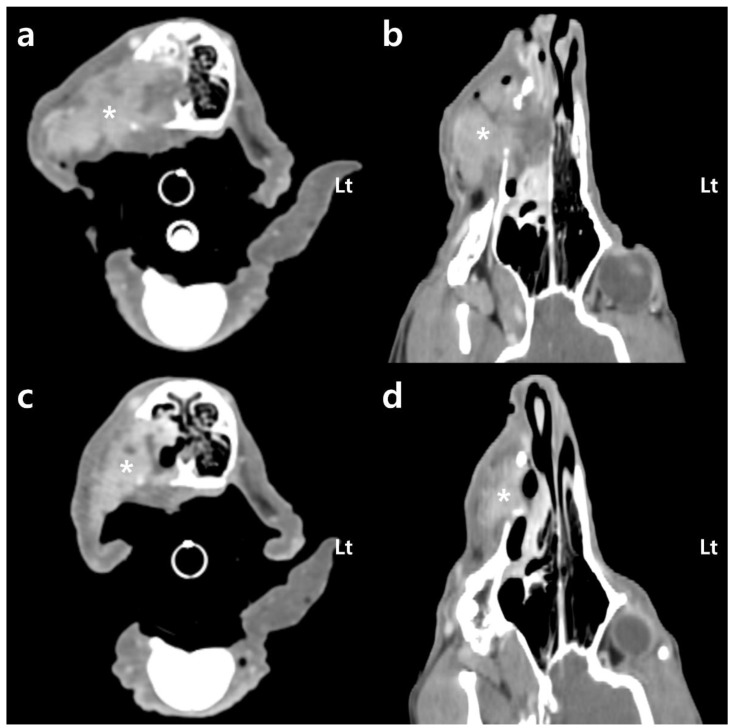
Transverse (**a**,**c**) and dorsal (**b**,**d**) multiplanar post-contrast computed tomography (CT) images of the patient with a soft tissue algorithm (WL 45; WW 450): (**a**,**b**) before radiotherapy, the right maxillary mass (*) reveals heterogeneous enhancement with a maximum diameter of 41 mm. Oral mass shows osteolysis of surrounding bones (maxillary bone, turbinate bone, and hard palate) and invasion into the right nasal cavity; (**c**,**d**) immediately after the last radiotherapy session, recognize that the size of the right maxillary mass decreased. When comparing the sum of the maximum diameters, the mass exhibits a 20.6% reduction in size. Lt: left.

**Figure 3 animals-14-01746-f003:**
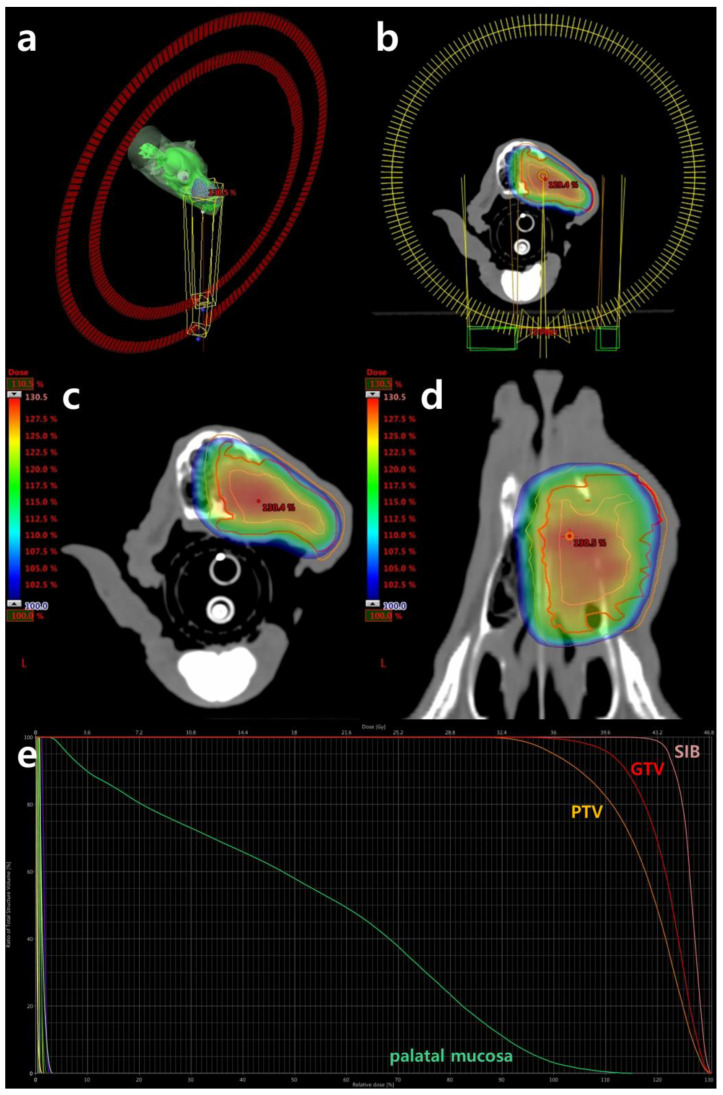
Radiation planning computed tomography scans using Varian Eclipse^TM^ (version 13.7.33) planning software: (**a**,**b**) the arcs encircling the oral mass illustrate the trajectory of the 360° volumetric arcs employed for radiation delivery, with one arc moving clockwise and the other counterclockwise; (**c**,**d**) the figures display the simultaneous integrated boost (SIB, pink), gross tumor volume (GTV, red), and planning target volume (PTV, orange) superimposed with the estimated radiation dose represented in color wash. It exclusively delineates regions receiving the prescribed dose (36 gray) or higher, with the red dot denoting the maximum hot spot for this slice; (**e**) the dose–volume histogram delineates targets and organs at risk, including SIB (pink), GTV (red), PTV (orange), and palatal mucosa (green). The histogram indicates that 99% of the GTV and 95% of the PTV are covered by at least 36 gray.

**Figure 4 animals-14-01746-f004:**
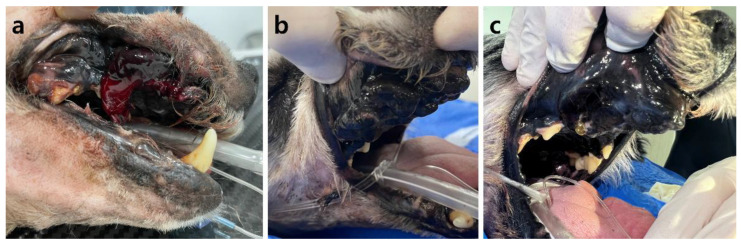
Gross images of oral melanoma: (**a**) before radiation therapy, the tumor exhibits continuous hemorrhage with blood clots adjacent to the tumor; (**b**) image after the second fraction demonstrates no further hemorrhage from the tumor; (**c**) image following the final treatment session reveals no additional hemorrhage from the tumor.

**Figure 5 animals-14-01746-f005:**
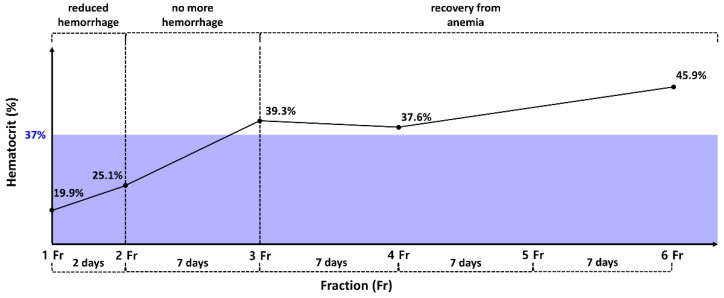
Changes in hematocrit (Hct) with respect to radiotherapy. Hct was measured immediately after each session of radiotherapy. Hemorrhage decreased after the first session and ceased after the second session. Hct increased as hemostasis was achieved through radiotherapy. Hct, which was 19.9% on the first day of treatment, increased to 25.1% after the second session when bleeding stopped, and, gradually, anemia recovered. The 37% on the y-axis represents the criterion for anemia.

**Table 1 animals-14-01746-t001:** Summary of dose and volume data.

	Volume	Dmax	Dmean	Dmin	D2%	D50%	D98%
GTV	17.8 cm^3^	46.9 Gy	43.1 Gy	28.0 Gy	46.4 Gy	43.8 Gy	36.5 Gy
PTV	26.2 cm^3^	46.9 Gy	42.3 Gy	27.3 Gy	46.3 Gy	43.0 Gy	34.4 Gy
Palatal mucosa	5.0 cm^3^	41.5 Gy	19.5 Gy	0.5 Gy	37.1 Gy	21.2 Gy	1.4 Gy
Eye (Rt)	5.1 cm^3^	1.1 Gy	0.4 Gy	0.2 Gy	0.9 Gy	0.3 Gy	0.2 Gy
Eye (Lt)	5.5 cm^3^	0.7 Gy	0.3 Gy	0.1 Gy	0.6 Gy	0.3 Gy	0.2 Gy
Lens (Rt)	0.6 cm^3^	0.9 Gy	0.6 Gy	0.3 Gy	0.8 Gy	0.5 Gy	0.3 Gy
Lens (Lt)	0.5 cm^3^	0.5 Gy	0.3 Gy	0.2 Gy	0.5 Gy	0.3 Gy	0.1 Gy
Brain	63.3 cm^3^	0.4 Gy	0.3 Gy	0 Gy	0.3 Gy	0 Gy	0 Gy
Optic nerve (Rt)	0.1 cm^3^	0.2 Gy	0.2 Gy	0.1 Gy	0.2 Gy	0.1 Gy	0 Gy
Optic nerve (Lt)	0.1 cm^3^	0.2 Gy	0.2 Gy	0.1 Gy	0.2 Gy	0.1 Gy	0 Gy
Optic chiasm	0.1 cm^3^	0.1 Gy	0.1 Gy	0.1 Gy	0.1 Gy	0 Gy	0 Gy

## Data Availability

The data supporting the findings of this study are available from the corresponding author upon request.

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
