# Peer review of "Application of Palliative Hemostatic Radiotherapy in Canine Unresectable Oral Melanoma: A Case Report"

_animals, 2024, doi:10.3390/ani14121746_

Round 1

Reviewer 1 Report

Comments and Suggestions for Authors

The case report entitled “Application of palliative hemostatic radiotherapy in canine oral
melanoma: A case report” is interesting and does add some to the literature.

I do have some points to address that I feel would increase the value of the case report. 

At anytime were clotting parameters assessed?

Additional staging information would be useful.

What is meant by Stage ¾ and then 4/4? Do you mean stage 3 and then stage 4?

Abstract

Line 24 – how was it assessed that the bleeding had reduced by 70%.

Case presentation

Lines 60-61 – What were other values on CBC to assess acute bleeding?  Reticulocyte count, TP.

Line 82 – I am assuming this included surrounding soft tissues of the maxilla?

Line 85 – 86 – How was this assessed?  CT? Thoracic radiographs?  Were lymph nodes aspirated – this is important especially since there was metastasis fairly quickly.

Line 103 – perhaps say medical palliative approach to make sentence more clear.

Line 109-110 – What was SIB dose plan?

Line 113-116 – Was there a CTV included and if not why?

Line 126 – What were other CBC parameters – ie retic count and TP?

Line 131 – Were lymph nodes sampled to confirm metastasis?  Was thoracic mets determined by CT or radiographs and were aspirates done?

Please provide dose statistics for the RT plan. Could see Rohrer Bley C, Meier VS, Besserer J, Schneider U. Intensity-modulated radiation therapy dose prescription and reporting: Sum and substance of the International Commission on Radiation Units and Measurements Report 83 for veterinary medicine. Vet Radiol Ultrasound. 2019 May;60(3):255-264. doi: 10.1111/vru.12722. Epub 2019 Feb 20. PMID: 30786324. For example.

Author Response

At anytime were clotting parameters assessed?

Revision) Unfortunately, we did not test aPTT (activated partial thromboplastin time) and PT (prothrombin time). As there were no bleedings observed in the patient apart from the bleeding from the oral tumor, the likelihood of bleeding due to a coagulation disorder was deemed low, thus coagulation tests were not performed.

What is meant by Stage ¾ and then 4/4? Do you mean stage 3 and then stage 4?

Revision) In the planning CT, the tumor size was assessed to be over 4cm, leading to a classification of WHO stage 3/4. However, on the CT scan immediately after the completion of radiation therapy, strong suspicion of regional lymph node and lung metastasis was observed, resulting in a classification of stage 4/4. Therefore, I wanted to express that the WHO stage had deteriorated from 3/4 to 4/4.

Line 24 – how was it assessed that the bleeding had reduced by 70%.

Revision)  I subjectively evaluated the reduction in bleeding, but using the quantitative term "70%" may not have been appropriate. I have revised the respective sentence in both the abstract and the main text.

Lines 60-61 – What were other values on CBC to assess acute bleeding?  Reticulocyte count, TP.

Revision)  I have included additional information below because it seemed that there was insufficient evidence to diagnose hemorrhagic anemia. At the time of presentation, the patient had an oral tumor accompanied by bleeding. Blood test results indicated anemia, hypochromia, macrocytosis, reticulocytosis, and hypoalbuminemia, leading to the diagnosis of hemorrhagic anemia with regenerative response. Detailed information regarding the blood tests is provided below. The main text has been supplemented with the blood test information indicating reticulocytosis.

visual inspection ⇒ oral mass with bleeding

HCT (28.6%, ref; 37~55) ⇒ anemia

MCV (82 fL, ref; 60~74 fL) ⇒ macrocytic

MCHC (30.2 g/dL, ref; 31~36 g/dL) ⇒ hypochromic

reticulocyte (4.26%, ref; 0.1~2.0%) ⇒ reticulocytosis

TP (5.7 g/dL, ref; 4.9~7.2 g/dL)

albumin (2.1 g/dL, ref; 2.3~3.9 g/dL) ⇒ hypoalbuminemia

globulin (3.6 g/dL, ref; 2.7~4.4 g/dL)

Line 82 – I am assuming this included surrounding soft tissues of the maxilla?

Revision) I have revised the expression as the original sentence seemed ambiguous. The intended message was to convey that bone resorption was observed in the bones adjacent to the oral tumor (maxillary bone, turbinate bone, hard palate).

Line 85 – 86 – How was this assessed?  CT? Thoracic radiographs?  Were lymph nodes aspirated – this is important especially since there was metastasis fairly quickly.

Revision) I have added information about the determination of the WHO stage through CT imaging. In canine oral melanoma, stage 3/4 can be diagnosed if the primary tumor size exceeds 40 mm regardless of the presence of metastasis in the regional lymph nodes. Distant metastasis was not identified on CT imaging. Although cytology/histology examinations were not performed on the regional lymph nodes, the absence of lymphadenopathy on CT and the lack of clear changes indicative of metastasis in the lymph nodes led to the conclusion that the likelihood of metastasis was low.

Line 103 – perhaps say medical palliative approach to make sentence more clear.

Revision) I have revised the sentence to clarify the expression

Line 109-110 – What was SIB dose plan?

Revision) I apologize for not specifying the dose for the SIB. We intended to deliver a dose of 120% of the prescription dose with the SIB.

Line 113-116 – Was there a CTV included and if not why?

Revision) As you mentioned, in cases other than SRT, it is common to consider microscopic disease and use CTV. However, in this case, due to the severe anemia, we simplified the planning by using only GTV and PTV to initiate radiation therapy as quickly as possible. Additionally, since it was radiation therapy for oral tumors, CTV was not used to reduce the risk of oral mucosal toxicity. Upon reflection, in similar cases of oral tumor patients at our hospital, when conducting palliative radiotherapy with larger fraction sizes (6 Gy/Fr or 8 Gy/Fr), acute toxicity was much rarer than expected. If treating the same patient again, we would likely decide to include CTV to enhance treatment effectiveness.

Line 126 – What were other CBC parameters – ie retic count and TP?

Revision) At the time of resolution of anemia, the following blood test results were obtained, in addition to hematocrit:

after 3rd fraction

after 4th fraction

visual inspection ⇒ no more hemorrhage

HCT (39.3%, ref; 37~55) ⇒ no more anemia

MCV (77 fL, ref; 60~74 fL) ⇒ macrocytic

MCHC (28.2 g/dL, ref; 31~36 g/dL) ⇒ hypochromic

reticulocyte (2.73%, ref; 0.1~2.0%) ⇒ reticulocytosis

visual inspection ⇒ no more hemorrhage

HCT (37.6%, ref; 37~55) ⇒ no more anemia

MCV (69.7 fL, ref; 60~74 fL) ⇒ no moremacrocytic

MCHC (33.3 g/dL, ref; 31~36 g/dL) ⇒ no more hypochromic

reticulocyte (0.1%, ref; 0.1~2.0%) ⇒ no more reticulocytosis

TP (6.6 g/dL, ref; 4.9~7.2 g/dL)

albumin (2.5 g/dL, ref; 2.3~3.9 g/dL) ⇒ no more hypoalbuminemia

globulin (4.1 g/dL, ref; 2.7~4.4 g/dL)

Line 131 – Were lymph nodes sampled to confirm metastasis?  Was thoracic mets determined by CT or radiographs and were aspirates done?

Revision) I see, thank you for the clarification. Here's the revised expression: "However, metastatic changes were strongly suspected in the right mandibular lymph node and lung on CT imaging, rather than definitively identified, leading to a worsening of the WHO stage to 4/4."

Please provide dose statistics for the RT plan. Could see Rohrer Bley C, Meier VS, Besserer J, Schneider U. Intensity-modulated radiation therapy dose prescription and reporting: Sum and substance of the International Commission on Radiation Units and Measurements Report 83 for veterinary medicine. Vet Radiol Ultrasound. 2019 May;60(3):255-264. doi: 10.1111/vru.12722. Epub 2019 Feb 20. PMID: 30786324. For example.

Volume

D50%

D98%

D2%

Conformity index

GTV

17.8 cm3

43.8 Gy

36.5 Gy

46.4 Gy

PTV

26.2 cm3

43.0 Gy

34.4 Gy

46.3 Gy

1.25

Volume

Dmean

D2%

Palatal mucsoa

5.0 cm3

19.5 Gy

37.1 Gy

Eye (Rt)

5.1 cm3

0.4 Gy

0.9 Gy

Eye (Lt)

5.5 cm3

0.3 Gy

0.6 Gy

Lens (Rt)

0.6 cm3

0.6 Gy

0.8 Gy

Lens (Lt)

0.5 cm3

0.3 Gy

0.5 Gy

Brain

63.3 cm3

0 Gy

0.3 Gy

Optic nerve (Rt)

0.1 cm3

0.2 Gy

0.2 Gy

Optic nerve (Lt)

0.1 cm3

0.2 Gy

0.2 Gy

Optic chiasm

0.1 cm3

0.1 Gy

0.1 Gy

Reviewer 2 Report

Comments and Suggestions for Authors

This is an informative case report showing the efficacy of palliative hemostatic radiotherapy in bleeding control of canine oral melanoma. Overall, I do not have major concern but have two questions for the authors.

(1) Since the tumor shrinkage was observed, was it possible that hemostatic effect was secondary to the tumor size reduction? which might lead to decreasing the pressure on surrounding blood vessels.

(2) Did author observed the recurrence of hemorrhage before patient's end.

Author Response

1) Since the tumor shrinkage was observed, was it possible that hemostatic effect was secondary to the tumor size reduction? which might lead to decreasing the pressure on surrounding blood vessels.

Revision) As you suggested, the reduction in tumor size may have contributed to the hemostatic effect of radiotherapy. The precise mechanism of the hemostatic effect of radiotherapy has not yet been elucidated, so we have discussed the proposed mechanisms in the discussion section.

2) Did author observed the recurrence of hemorrhage before patient's end.

Revision) Fortunately, in this case, there was no recurrence of bleeding after it ceased. However, in a comparative study of palliative radiotherapy protocols for hemostasis in human medicine, it was reported that bleeding recurred in 25% (22/88) of cases within a median of 84 days after hemostasis.

Reviewer 3 Report

Comments and Suggestions for Authors

Dear authors

Despite the innovation in this case report, some changes are necessary.

As this is a case report with only 1 case, the authors should be more careful with the way they write. Use expressions such as "may be beneficial for dogs with..." instead of "has good results, is a good therapy". In reality, little can be concluded from a single clinical case. 

I have some concerns about the diagnosis of this neoplasm. The definitive diagnosis and especially the malignant characteristics should be guided by a histopathological analysis according to WHO criteria. Cytology is not enough. The authors do not present results regarding the number of mitoses per 10HPF, nuclear grid, vascular tumor emboli, etc... none of the histological criteria for malignancy were evaluated. 

A paragraph with the limitations of the study and the need for future studies should be included in the discussion. 

The conclusions are diretioned to human medicine and saying that this therapy is a good option. However, we cannot say this for veterinary medicine. Reword the conclusions, directing them only to this clinical case and not generalizing.

Author Response

As this is a case report with only 1 case, the authors should be more careful with the way they write. Use expressions such as "may be beneficial for dogs with..." instead of "has good results, is a good therapy". In reality, little can be concluded from a single clinical case. 

Revision) As you suggested, despite the limited number of cases in animals, there seems to be confidence in the hemostatic effects. Therefore, I have revised the discussion and conclusion to use "may be beneficial" to express the hemostatic effects.

I have some concerns about the diagnosis of this neoplasm. The definitive diagnosis and especially the malignant characteristics should be guided by a histopathological analysis according to WHO criteria. Cytology is not enough. The authors do not present results regarding the number of mitoses per 10HPF, nuclear grid, vascular tumor emboli, etc... none of the histological criteria for malignancy were evaluated. 

Revision) As you mentioned, histopathological examination is the most accurate diagnostic method. However, if melanin pigmentation is present, diagnosis can be straightforward based on cytological examination alone. In this case, cytological examination of the oral tumor revealed immature mesenchymal cells with low cellular differentiation, accompanied by numerous black-green granules in the cytoplasm, allowing for a diagnosis of malignant melanoma.

A paragraph with the limitations of the study and the need for future studies should be included in the discussion.

Revision) It seems that the discussion section lacked content regarding limitations and further studies. I have added additional information to the discussion section concerning limitations and the need for further studies.

The conclusions are diretioned to human medicine and saying that this therapy is a good option. However, we cannot say this for veterinary medicine. Reword the conclusions, directing them only to this clinical case and not generalizing.

Revision) As you suggested, despite the limited number of cases in animals, there seems to be confidence in the hemostatic effects. Therefore, I have revised the discussion and conclusion to use "may be beneficial" to express the hemostatic effects.

Reviewer 4 Report

Comments and Suggestions for Authors

In this case report the authors describe the treatment of one 9-year old Schnauzer with an oral melanoma with palliative radiotherapy for hemostasis. 

After reading this manuscript I returned to the abstract and found the abstract to be rather misleading. The hematocrit level prior to VMAT was 19.9% this was an improvement from 18.2% prior to transfusion (line 21). I suggest the abstract be rewritten to focus on the effect of the radiotherapy not the transfusion. In line 24 and 25 the authors indicated that the hematocrit "increased, resulting in anemia". I assume that they mean that anemia resolved, otherwise please clarify further. In the abstract and throughout the manuscript the authors discuss animal intake and pain management as measures of efficacy. There is however no data presented to support those claims. The authors should omit those sections of the manuscipt and focus on hemostasis or provide the data quantifying feed intake and pain scoring changes for this case. 

One of the values of case reports are when they educate partitioners on novel treatment options. In those cases, it is especially helpful when the authors explain and justify the treatment parameters. It is not clear in this paper why 6 fractions of 6 Gy was selected. The authors do mention that articles that suggest 5 fractions and a BED of less than <39. And that fewer than five may be preferable, why did this patient have 6 fractions and why was 6 Gy selected. What suggestions would you offer for future cases, should they follow this same protocol or 8 Gy. 

Change foul odor to halitosis in line 19

Was the patient on any other medication over the course of this treatment? It sounds like there was at least some additional pain medication given over the 1st few weeks of treatment. That should be included and may confound these results. 

Line 127 how was intake quantified and how was pain determined. 

Strike the last sentence in line 155-56 and line 167-68 or provide evidence to support these claims.

The font and axis labels on Figure 3 are difficult to read and should be enlarged.  

Was blood work other than hematocit checked again in this patient and were there changes in serum biochemistry in response to radiotherapy. 

Figure 5: Please clarify if these values were obtained prior to or after each radiotherapy session. 

Strike line last sentence from the conclusion or include pain and quality of life data. 

 There is a lack of consistency in the citing of references. Remember to double check even when using reference software. Some of the journals are abbreviated some are not. Some of the article title have capital letter throughout and some only at the beginning. This should be standardized and meet journal criteria.  

Overall, this is an interesting case study, please provide the justification for the administered treatment and be careful about making claims and statements without providing data to back those up. 

Author Response

Line 21 - After reading this manuscript I returned to the abstract and found the abstract to be rather misleading. The hematocrit level prior to VMAT was 19.9% this was an improvement from 18.2% prior to transfusion (line 21). I suggest the abstract be rewritten to focus on the effect of the radiotherapy not the transfusion.

Revision) As you suggested, it may be appropriate to include the Hct value (19.9%) before VMAT to emphasize the hemostatic effect of radiotherapy. However, I chose to express it this way to highlight the cessation of bleeding and the resolution of anemia following palliative radiotherapy, given the severity of the hemorrhagic anemia requiring transfusion.

Line 24 & 25 - In line 24 and 25 the authors indicated that the hematocrit "increased, resulting in anemia". I assume that they mean that anemia resolved, otherwise please clarify further.

Revision) I misspoke and have corrected the expression. "Resolution of anemia following radiotherapy" is the accurate phrase.

In the abstract and throughout the manuscript the authors discuss animal intake and pain management as measures of efficacy. There is however no data presented to support those claims. The authors should omit those sections of the manuscipt and focus on hemostasis or provide the data quantifying feed intake and pain scoring changes for this case. Was the patient on any other medication over the course of this treatment? It sounds like there was at least some additional pain medication given over the 1st few weeks of treatment. That should be included and may confound these results.

Line 127 how was intake quantified and how was pain determined. 

Revision) As you mentioned, the previous version lacked objective criteria for pain assessment and details on analgesic use. Therefore, we have added this information. For pain evaluation, we used the "Colorado State University Veterinary Medical Center Canine Acute Pain Scale." Before radiation therapy, despite the use of various analgesics (piroxicam, gabapentin, tramadol), the patient exhibited a pronounced pain response upon palpation. However, from the fourth session of radiation therapy onwards, there was minimal pain response upon palpation, even without the use of analgesics. According to the pain response scoring criteria (Colorado State University Canine Acute Pain Scale), improvement was noted from "pre-radiotherapy (score 3/4)" to "post-radiotherapy (1/4)".

cf) Colorado State University Canine Acute Pain Scale : CSU_VTH_Canine_Acute_Pain_Scale (colostate.edu)

Although food intake was not quantitatively assessed, it was observed that the patient, who was barely eating before radiation therapy, showed a gradual increase in food consumption starting from the second session.

One of the values of case reports are when they educate partitioners on novel treatment options. In those cases, it is especially helpful when the authors explain and justify the treatment parameters. It is not clear in this paper why 6 fractions of 6 Gy was selected. The authors do mention that articles that suggest 5 fractions and a BED of less than <39. And that fewer than five may be preferable, why did this patient have 6 fractions and why was 6 Gy selected. What suggestions would you offer for future cases, should they follow this same protocol or 8 Gy. 

Revision) As you mentioned, there seemed to be insufficient justification for selecting 6 Gy, 6 Fr in the text. If the goal was solely the hemostatic effect in hemorrhagic tumors, "5 fractions or less" or "8 Gy, single fraction" might have been more appropriate. However, in this case, the chosen palliative radiotherapy protocol commonly used for canine oral melanoma (36 Gy = 6 Gy x 6 Fr, weekly) was employed not only for its hemostatic effect but also for its palliative benefits.

Change foul odor to halitosis in line 19

Revision) I have revised the manuscript to correct inappropriate terminology.

Strike the last sentence in line 155-56 and line 167-68 or provide evidence to support these claims.

Revision) Line 155-156 : I remove the content of lines 155-156 as it is not visible in the figure.

Revision) Line 167~168 : In terms of pain management, as mentioned in the previous revision, prior to radiotherapy, the pain response upon palpation was severe, and pain was unmanageable even with analgesics. However, after the fourth treatment session, pain response upon palpation was minimal. Regarding odor, before radiotherapy, there was a severe halitosis detected during oral examination. However, as the radiotherapy sessions progressed, a decrease in halitosis was observed.

The font and axis labels on Figure 3 are difficult to read and should be enlarged.

Revision) Unfortunately, Figure 3 was captured from the radiation therapy planning program (ECLIPSE, Varian), and adjusting the font size within the program was not feasible. However, the information regarding the x and y axes is as follows. The x-axis represents relative dose (%), while the y-axis represents volume ratio (%).

Was blood work other than hematocit checked again in this patient and were there changes in serum biochemistry in response to radiotherapy. 

Revision) Prior to radiotherapy, CRP levels were elevated, but after radiotherapy, they decreased to within the reference range. Therefore, this suggests a reduction in inflammation following radiation therapy

pre-RT : CRP 17.1 mg/L (ref; 0~10.0 mg/L)

post-RT : CRP 1.2 mg/L (ref; 0~10.0 mg/L)

Revision) At the time of resolution of anemia, the following blood test results were obtained, in addition to hematocrit:

after 3rd fraction

after 4th fraction

visual inspection ⇒ no more hemorrhage

HCT (39.3%, ref; 37~55) ⇒ no more anemia

MCV (77 fL, ref; 60~74 fL) ⇒ macrocytic

MCHC (28.2 g/dL, ref; 31~36 g/dL) ⇒ hypochromic

reticulocyte (2.73%, ref; 0.1~2.0%) ⇒ reticulocytosis

visual inspection ⇒ no more hemorrhage

HCT (37.6%, ref; 37~55) ⇒ no more anemia

MCV (69.7 fL, ref; 60~74 fL) ⇒ no moremacrocytic

MCHC (33.3 g/dL, ref; 31~36 g/dL) ⇒ no more hypochromic

reticulocyte (0.1%, ref; 0.1~2.0%) ⇒ no more reticulocytosis

TP (6.6 g/dL, ref; 4.9~7.2 g/dL)

albumin (2.5 g/dL, ref; 2.3~3.9 g/dL) ⇒ no more hypoalbuminemia

globulin (4.1 g/dL, ref; 2.7~4.4 g/dL)

Figure 5: Please clarify if these values were obtained prior to or after each radiotherapy session. 

Revision) The hematocrit presented in the figure 5 represents values measured after each session of radiation therapy.

Strike line last sentence from the conclusion or include pain and quality of life data. 

Revision) The discussion regarding pain has been mentioned in the previous revision, and it was assumed that an improvement in pain naturally leads to an increase in quality of life.

 There is a lack of consistency in the citing of references. Remember to double check even when using reference software. Some of the journals are abbreviated some are not. Some of the article title have capital letter throughout and some only at the beginning. This should be standardized and meet journal criteria.  

Revision) As you suggested, I have corrected the inconsistency in my reference citations for uniformity.

Round 2

Reviewer 1 Report

Comments and Suggestions for Authors

Thank you for addressing my concerns in the manuscript. My only suggestion is that the table you provided of the radiation plan information would be best included as a table in the manuscript.

Author Response

My only suggestion is that the table you provided of the radiation plan information would be best included as a table in the manuscript.

Revision) We have newly added a table to the manuscript with information on the volume and dose of the structures.

Reviewer 3 Report

Comments and Suggestions for Authors

The authors mentioned that  "if melanin pigmentation is present, diagnosis can be straightforward based on cytological examination alone". In fact it is, but we can only say that is an melanoma and not an malignant melanoma. The authors also refer that "cytological examination of the oral tumor revealed immature mesenchymal cells with low cellular differentiation, accompanied by numerous black-green granules in the cytoplasm, allowing for a diagnosis of malignant melanoma.". This criteria is not enough to classify as an MALIGNANT melanoma. I advise the authors to study some current research with histological criteria, such as https://doi.org/10.3389/fvets.2024.1359426     and     https://doi.org/10.3390/vetsci9040175

Author Response

The authors mentioned that  "if melanin pigmentation is present, diagnosis can be straightforward based on cytological examination alone".  In fact it is, but we can only say that is an melanoma and not an malignant melanoma. The authors also refer that "cytological examination of the oral tumor revealed immature mesenchymal cells with low cellular differentiation, accompanied by numerous black-green granules in the cytoplasm, allowing for a diagnosis of malignant melanoma.". This criteria is not enough to classify as an MALIGNANT melanoma. I advise the authors to study some current research with histological criteria, such as https://doi.org/10.3389/fvets.2024.1359426 and https://doi.org/10.3390/vetsci9040175

Revision) Thank you for your thorough review of my paper. Your insights have been invaluable in enhancing the quality of my work. As you pointed out, it seems incorrect to confirm a diagnosis of malignant melanoma based solely on cytological examination in this case. Therefore, we have removed the statement that malignant melanoma was confirmed through cytological examination from the manuscript. Instead, we have revised it to state that immature mesenchymal cells with melanin pigmentation were identified, along with features indicative of high cellular malignancy such as prominent multiple nuclei, anisocytosis, anisokaryosis, and variation in the nucleus-to-cytoplasm ratio. Based on these findings, the oral mass was provisionally diagnosed as malignant melanoma.

Reviewer 4 Report

Comments and Suggestions for Authors

I want to thank the authors for their prompt resubmission and responses to all of my reviewer comments. I want to apologize for any confusions, my comments were meant to identify areas that needed revising and clarification in the manuscript. These issues and points of confusion I assume our colleagues might share. Rather than just providing me with this information, I would like to see these details incorporated into the revised manuscript. Although the reviewers did start each response with the term "revision" it would seem that not all of these items were addressed in the paper, nor was there always clear the justification why they couldn't be made. Deciding not to use the HCT prior to radio therapy but rather an earlier lower value (for example) was not well supported. I am willing to continue to review this interesting paper and look forward to reading the next revision with these details added. 

Author Response

Thank you for your thorough review of my paper. Your insights have been invaluable in enhancing the quality of my work. I have carefully noted the points you raised, and where applicable, revisions have been made in the manuscript as indicated by "revision." For areas where no revisions were made, I have provided explanations. If you have any further suggestions or feedback, I am more than willing to incorporate them. Thank you once again for your time and valuable input.

Line 21 - After reading this manuscript I returned to the abstract and found the abstract to be rather misleading. The hematocrit level prior to VMAT was 19.9% this was an improvement from 18.2% prior to transfusion (line 21). I suggest the abstract be rewritten to focus on the effect of the radiotherapy not the transfusion.

Explanation) At the time of the planning CT for radiotherapy, the patient showed anemia with an Hct of 28.6%. Typically, radiotherapy planning takes up to 7 days. Four days after the planning CT was taken, the Hct dropped to 18.2%, necessitating a blood transfusion, which temporarily increased the Hct to 23.3%. However, right after the first radiotherapy session, the Hct decreased again to 19.9%.

2023.08.09 (planning CT): Hct 28.6%

2023.08.13: Hct 18.2% => blood transfusion needed

2023.08.15: Hct 23.3% => temporary increase

2023.08.16 (1st fraction): Hct 19.9% => still low HCT

The rationale for using the Hct value of 18.2% (on the day of the planning CT for radiotherapy) instead of 19.9% (on the first day of radiotherapy) in the abstract is as follows:

  1. The Hct of 19.9% was measured right after the first radiotherapy session, so it might have been slightly affected by the treatment.
  2. The Hct of 18.2% was obtained on the day of the planning CT, the point at which the decision to start radiotherapy was made. Both Hct values, 18.2% and 19.9%, fall within the radiotherapy period. We wanted to emphasize that the hemorrhagic anemia, which could not be resolved by transfusion, was alleviated by hemostasis through radiotherapy. Therefore, we used the lower Hct value (18.2%) in the abstract.

Was blood work other than hematocit checked again in this patient and were there changes in serum biochemistry in response to radiotherapy. 

Revision) We have also newly added information on MCH, MCHC, and CRP before and after radiotherapy to the manuscript, in addition to Hct and reticulocytosis.

Figure 5: Please clarify if these values were obtained prior to or after each radiotherapy session. 

Revision) We have newly added to the manuscript that the Hct was measured immediately after each session of radiotherapy.

Round 3

Reviewer 3 Report

Comments and Suggestions for Authors

thanks for the alterations

Reviewer 4 Report

Comments and Suggestions for Authors

Thank you for the continued work to improve this manuscript and clarify points of confusion. I still note inconsistency in reference formatting, but otherwise the manuscript is publishable.

Author Response

It appears that the references are not yet fully organized. I will revise them accordingly. Thank you.